# Mediating Effect of Psychological Process Variables on the Relationship between Dysfunctional Coping and Psychopathologies: A Comparative Study on Psychopathologies during COVID-19

**DOI:** 10.3390/bs12070206

**Published:** 2022-06-24

**Authors:** Nurfarah Lydia Hambali, Friska Ayu, Nicholas Tze Ping Pang, Mohd Amiruddin Mohd Kassim, Hafid Algristian, Moch. Sahri, Nelbon Giloi, Syed Sharizman Syed Abdul Rahim, Azizan Omar, Mohammad Saffree Jeffree, Walton Wider

**Affiliations:** 1Faculty of Medicine and Health Sciences, Universiti Malaysia Sabah, Kota Kinabalu 88400, Malaysia; nurfarahlydia@ums.edu.my (N.L.H.); amiruddink@ums.edu.my (M.A.M.K.); nelbon.giloi@ums.edu.my (N.G.); syedsharizman@ums.edu.my (S.S.S.A.R.); azizan.omar@ums.edu.my (A.O.); saffree@ums.edu.my (M.S.J.); 2Faculty of Health, Universitas Nahdlatul Ulama Surabaya (UNUSA), Surabaya 60237, Indonesia; friskayuligoy@unusa.ac.id (F.A.); sahrimoses@unusa.ac.id (M.S.); 3Faculty of Medicine, Universitas Nahdlatul Ulama Surabaya (UNUSA), Surabaya 60237, Indonesia; dr.hafid@unusa.ac.id; 4Faculty of Business and Communication, INTI International University, Nilai 71800, Malaysia

**Keywords:** COVID-19, Indonesia, Malaysia, psychological mindedness, psychological inflexibility, psychological mindfulness, psychopathologies, dysfunctional coping

## Abstract

The COVID-19 crisis has had repercussions on global mental wellbeing. This study aimed: (1) to identify the mediating role of psychological process variables, namely psychological mindedness, psychological mindfulness, and psychological inflexibility on the relationship between dysfunctional coping and psychopathologies in Indonesian undergraduate students subjected to national quarantine orders throughout July, 2020 and (2) to compare the level of anxiety, depression, and anxiety between Indonesian and Malaysian undergraduate students. A cross-sectional study was performed with 869 Indonesian undergraduate students from Nahdlatul Ulama University of Surabaya (UNUSA) and 515 undergraduate students from Universiti Malaysia Sabah (UMS). The BIPM, MAAS, AAQ-I, DASS-21, and Brief COPE were used to assess the research variables. The proportion who scored “moderate” and above for depression, anxiety, and stress were 20.2%, 25.0%, and 14.2%, respectively, in Malaysian samples and 22.2%, 35.0%, and 23.48% in Indonesian samples. In Study 1, psychological mindedness, psychological mindfulness, and psychological inflexibility significantly mediated the relationship between dysfunctional coping and psychopathologies. In Study 2, Indonesians demonstrated significantly higher anxiety and stress compared to Malaysian samples. Despite the contrasting COVID-19 situations in Malaysia and Indonesia, psychopathologies were more affected in Indonesia. Hence, our study suggests how crucial it is for mental health providers to consider promoting psychological mindedness, psychological mindfulness, and psychological flexibility to alleviate the corresponding psychopathologies among undergraduate students.

## 1. Introduction

Coronavirus disease (COVID-19), a novel coronavirus which emerged from Wuhan, China in late 2019, was declared a pandemic on 12 March 2020 by the World Health Organization (WHO). The pandemic reached Malaysia in January 2020 from travelers in China arriving via Singapore on 25 January 2020. With the escalation of active cases in March 2020 especially after the emergence of localised clusters, the Malaysian government implemented a Movement Control Order (MCO) from 18 March 2020. The MCO comprised of restrictions on mass gathering, travelling abroad, closure of all government and private schools and universities, and closure of all government and private premises except for essential services [1,2].

Likewise, in Indonesia, Jakarta was the first to implement large-scale social restrictions, also known as *Pembatasan Sosial Berskala Besar* (PSBB) starting from 10 April 2020 [3]. The Nahdlatul Ulama University of Surabaya (UNUSA) in Surabaya, East Java province, implemented the study from home policy on 22 March 2020 before Surabaya implemented the PSBB on 26 April 2020. Many universities were forced to shift towards e-learning, which relies disproportionately on university computer lab services, poor internet connections, and excessive consumption of Internet data for video conferences. Literature suggests that college and university students were stressed about dormitory evacuation, cancellation of planned events, loss of part-time jobs, anxiety about the job market, and the potential for abuse at home [4]. Furthermore, COVID-19 negatively affected undergraduate students’ mental health in terms of stress, disappointment, loneliness, financial setbacks, and relocation [5]. As asserted by Sun et al. [6], university students are considered a group of people who are vulnerable to distress and mental health disorders during COVID-19. Thus, it is pertinent to determine the risks and protective factors towards students’ mental health during the pandemic.

Many factors can contribute to students’ psychological distress, and analysing such underlying psychological process variables would assist in a better understanding of particular psychopathologies [7,8]. Three psychological process variables were explored in this study, namely psychological mindedness, psychological mindfulness, and psychological inflexibility. Psychological mindedness refers to the intrinsic motivation to be in touch with one’s inner feelings and thoughts by monitoring and analysing them in an adaptive way [9,10]. For psychological mindfulness, there are two main elements of mindfulness in a clinical psychology perspective: the awareness of one’s moment-to-moment experience nonjudgmentally and with acceptance [11]. Psychological inflexibility is a process wherein an individual depicts dysfunctional control endeavors connected with real sensations, feelings, or thoughts to try not to encounter upsetting internal or external events [12]. Studies have shown significant correlations between psychological mindedness, psychological mindfulness, psychological inflexibility, and psychopathologies. For example, psychological mindedness was found to be inversely related to symptoms of psychological distress and psychopathologies [13,14]. Meanwhile, low levels of mindfulness were found to be associated with depression, anxiety, neuroticism, dissociation, insecure attachment, negative affect, and difficulties in emotion regulation [15,16,17,18]. Likewise, extensive evidence has shown that psychological flexibility is positively associated with psychological well-being and inversely associated with a wide range of distress, including depression, anxiety, and general psychological distress [19,20,21,22,23].

Malaysia and Indonesia adopted fairly strict social restrictions to prevent COVID-19 transmission, resulting in large-scale quarantines and isolation. It has been known that quarantine is a risk factor for mental health issues, with longer quarantines associated with poorer psychological outcomes [24]. However, despite the measures taken to flatten the curve, there was a stark difference between the two countries. Indonesia was among the worst hit countries in the world, while Malaysia was able to handle COVID-19 relatively well. This study was hence performed to assess the effect on underlying psychological processes, namely psychological mindedness, psychological mindfulness, and psychological inflexibility, on the relationship between dysfunctional coping and psychopathologies during the COVID-19 pandemic. Moreover, it aims to determine if the difference of the COVID-19 burden in these two countries would affect the level of anxiety, depression, and stress of the respective populations.

## 2. Methods

### 2.1. Study Design and Participants

This study used a cross-sectional design by utilising survey methods to investigate the relationship between psychological mindedness, psychological mindfulness, psychological inflexibility, dysfunctional coping, and psychopathologies.

The study population from Indonesia encompassed university students in UNUSA located at Surabaya, East Java province, who were subject to the large-scale social restrictions (PSBB). As UNUSA did not have on-campus hostels, hence, respondents were not restricted within the university campus. However, they were still subjected to the implementation of PSBB in their respective provinces. The students were either staying inside Surabaya or outside of Surabaya. On the other hand, the study population from Malaysia encompassed university students in three campuses of a public university in East Malaysia (Sabah) who were subject to the nationwide Movement Control Order. All respondents were restricted to movement within the university campus only. All respondents were aged within 17–30 years old.

### 2.2. Ethical Considerations

The study was conducted according to the guidelines of the Declaration of Helsinki and approved by the medical ethics review committee of Universiti Malaysia Sabah (JKetika 2/20 (2), 27 April 2020).

### 2.3. Research Instruments

Firstly, basic sociodemographic characteristics were assessed in both groups. Secondly, in the Indonesian students, five scales were used: the DASS-21 measuring depression, anxiety, and stress (psychopathologies); the Balanced Index of Psychological Mindedness (BIPM) for psychological mindedness; the Acceptance and Action Questionnaire-II (AAQ-II) for psychological inflexibility; Coping Orientation to Problems Experienced Inventory (Brief COPE) Inventory for dysfunctional coping; and the Mindfulness Attention and Awareness Scale (MAAS) for psychological mindfulness. The DASS-21 has been validated both in the Malay and Indonesian language. The BIPM, AAQ-II, and MAAS are only available in Malay language validations. However, Malay and Indonesian languages are mutually intelligible; hence, the lead researchers from Indonesia made minor adjustments to all three Malay scales in order to ensure full intelligibility. Due to time pressures in the dynamic urgency of COVID-19, there was insufficient time to perform a full validation. For the Malaysian students, data were extracted from a separate project in a similar cohort of university students where only DASS-21 scores were measured.

#### 2.3.1. The Depression, Anxiety and Stress Scale—21 Items (DASS-21)

The DASS-21 questionnaire is a set of scale consisting of 21 items measuring the emotional states of depression, anxiety, and stress [25]. Respondents were requested to complete the DASS-21 based on the presence of a symptom over the previous week. They rated each item on a four-point scale ranging from 1 (*did not apply to me at all over the last week*) to 4 (*applied to me very much or most of the time over the past week*), with higher scores indicating greater severity. The Malay version of DASS-21 has reasonable internal consistency with Cronbach’s alpha values ranges from 0.74 to 0.84 [26].

#### 2.3.2. Balanced Index of Psychological Mindedness (BIPM)

The BIPM is a self-report psychological instrument that measures psychological mindedness and was developed by [27]. It comprises of 14 items and 2 factors: Interest and Insight. *Interest* refers to attending to one’s internal feelings, and *Insight* refers to understanding these feelings [28,29]. Each item is rated on a five-point Likert scale ranging from 1 (not true) to 5 (very much true). The Interest and Insight subscales of the BIPM showed good internal consistency (Cronbach’s alphas = 0.85 and 0.76, respectively), test-retest reliability (r = 0.63 and 0.71, respectively), and construct validity (e.g., r > 0.40 with related constructs). The Malay version of the BIPM was preliminarily validated and has a Cronbach’s alpha for total scores of 0.79, as well as for the “Interest” (0.76) and “Insight” (0.75) subscales, which are acceptable [30,31].

#### 2.3.3. Acceptance and Action Questionnaire-II (AAQ-II)

The AAQ-II is an instrument to assess experiential avoidance and psychological inflexibility [32]. Experiential avoidance is defined as the attempt of the individual to ignore unpleasant thought, feelings, and physical sensations that leads to measures against one’s values, causing continuing harm. Meanwhile, psychological inflexibility refers to rigid psychological reaction against one’s value in order to avoid distress, uncomfortable feelings, and thought and tend to ignore the present moment [33]. AAQ-II is a unidimensional scale with 7 items and is rated based on 7-point scale. AAQ-II has good internal consistency (Cronbach’s alpha = 0.88) and good test retest reliability over 3 and 12 months at 0.81 and 0.79, respectively [32]. Higher AAQ-II scores reflected significant psychological inflexibility and were also found to be associated with greater levels of depressive symptoms, anxiety and stress, thought suppression, and psychological distress.

#### 2.3.4. Mindful Attention Awareness Scale (MAAS)

The MAAS measures the level of awareness and attention to the present-moment experience [16]. It consists of 15 items, and participants respond to each item using a six-point scale ranging from 1 (*almost always*) to 6 (*almost never*). Higher scores indicate higher levels of psychological mindfulness [34]. The Malay version of MAAS showed an acceptable internal consistency with Cronbach’s alpha of 0.851 [35].

#### 2.3.5. Coping Orientation to Problems Experienced Inventory (Brief COPE) Inventory

The Brief COPE is a self-report instrument containing 28 items that enquire regarding the frequency of various coping methods [36]. Individuals use a 4-point scale to respond regarding the frequency of specific coping methods, from 1 (*I usually don’t do this at all*) to 4 (*I usually do this a lot*). The scale yields scores on three factors: problem-oriented coping styles, emotion-oriented coping styles, and dysfunctional coping styles. For the entire Malay scale, the internal consistencies ranged from 0.51 to 0.99. The test–retest Intraclass Correlation Coefficient (ICC) ranged from < 0.00 to 0.98 [37]. For the purpose of this study, we only used the dysfunctional coping style.

### 2.4. Data Analysis

In Study 1, we applied partial least squares–structural equation modeling (PLS-SEM) to assess measurement models and structural models for the Indonesian sample. We utilized PLS-SEM because of the inborn appropriateness of this methodology for exploratory investigations [38]. In addition, the comparison of anxiety, depression, and stress among Indonesian and Malaysian samples was performed using SPSS Version 27.

### 2.5. Reliability and Validity

For the Indonesian sample, a total of 869 samples were used to evaluate the measurement and structural models. At first, the reliability and validity of the reflective constructs (psychological mindedness, psychological mindfulness, psychological inflexibility, dysfunctional coping, and psychopathologies) were assessed. In order to confirm reliability, the composite reliability (CR) should be higher than 0.7, whereas for convergent validity to be established, the average variance extracted (AVE) should be higher than 0.5 [38]. Twenty-five indicators were removed due to weak outer loadings. Table 1 presents a summary for all reflective constructs, demonstrating that reliability and convergent validity had been verified. Next, to ascertain the discriminant validity, we used the Fornell–Larcker criterion [39]. According to Table 2, the Fornell–Larcker results demonstrated that the square root of the AVE for each construct (bold) was higher than its correlation with other constructs; discriminant validity was therefore established.

## 3. Results

### 3.1. Descriptive Statistics

Table 3 shows the respondent profiles who participated in this study. Out of 869 Indonesian respondents, 736 (84.7%) were females. The mean age for Indonesian respondents was 20.5 years old. Out of 515 Malaysian respondents, 357 (69.3%) were females. The mean age for the Malaysian respondents was 22.2 years old. Furthermore, the descriptive analyses of DASS-21 revealed that the majority of Indonesian respondents had a normal level of depression (61.0%), whereas the majority of Malaysian respondents had mild levels of depression (38.1%). In terms of anxiety, the majority of Indonesian respondents had a normal level (46.3%), whereas the majority of Malaysian respondents had a mild level (41.7%). Last but not least, for stress, the majority of Indonesian respondents had a normal level (64.6%), whereas the majority of Malaysian respondents had moderate level (40.4%).

### 3.2. Study 1

Table 1 shows the mean scores and standard deviations (SD) for research variables for Study 1. The mean score for psychopathologies was 1.93 (SD = 0.65); 1.97 for dysfunctional coping (SD = 0.64); 3.84 for psychological mindfulness (SD = 1.03); 2.39 for psychological mindedness (SD = 0.88); and 3.35 for psychological inflexibility (SD = 1.39).

Table 4 also shows that all inner VIF values were below 5, therefore indicating that there was no issue of collinearity among the predictor constructs in the structural model [40]. To assess the direct effect and indirect effect, we assessed the structural model by computing the path coefficient (*β*), t-values, *p*-values, and *R^2^* using a 5000-sampling bootstrapping technique [40]. According to Table 4 and Figure 1, all direct effects were significant. With regard to the ∆R^2^ value, the findings showed that the ∆R^2^ value ranged from 0.091 to 0.515, which was acceptable [40] and accounted for 9.1% to 51.5% of the exploratory variance.

To assess the mediator in this study, the product of the coefficient approach using the bootstrapping resampling method was applied [41]. Our study showed that psychological mindedness, psychological mindfulness, and psychological inflexibility significantly mediate the relationship between dysfunctional coping and psychopathologies (refer Table 4).

### 3.3. Study 2

In Study 2, the information was obtained from secondary data source, namely a separate but similar project performed by a similar research group. Combining the Indonesian study with the Malaysian data, there was a total of 1382 respondents consisting of 291 males (21.0%) and 1091 females (78.8%). Both studies utilized students from undergraduate age groups. As per Table 5, there was a significant difference in anxiety and stress between both countries. Specifically, Indonesian participants (M = 1.78, SD = 0.55) compared to the Malaysian participants (M = 1.64, SD = 0.52) demonstrated significantly higher anxiety: t(1380) = 3.342, *p* = 0.001. In addition, Indonesian participants (M = 1.93, SD = 0.63) compared to the Malaysian participants (M = 1.82, SD = 0.59) demonstrated significantly higher stress: t(1380) = 4.509, *p* = 0.000. There were no differences in depressive symptoms between both countries.

## 4. Discussion and Conclusions

The COVID-19 pandemic has resulted in major impacts on many sectors such as economy, health, and education. With multiple nations imposing stay-at-home orders and lockdowns, mental wellbeing is paramount as there is a restriction of freedom alongside changes in routine. Many previous studies show the long term effects on people who were subjected to quarantine even months after the quarantine was over. This study aspires to understand the underlying psychological processes that cause people to either have a dysfunctional coping style or psychopathology symptoms, with the clinical implication of contributing to prospective interventions to mitigate emotional distress in individuals impacted by this pandemic.

The Indonesian data demonstrates that 22.2% of respondents scored “moderate” and above (severe and extremely severe) for depression, 35.0% scored “moderate” and above for anxiety, and 23.48% scored “moderate” and above for stress. This is also consistent with a previous study conducted in China in assessing the impact of COVID-19 on anxiety in Chinese university students [42]. Dysfunctional coping styles were significantly related to psychopathologies, which is consistent with previous studies [30,43]. The three psychological process variables, namely psychological mindedness, psychological mindfulness, and psychological inflexibility, completely mediated the relationship between dysfunctional coping mechanisms and psychopathologies. Specifically, dysfunctional coping is shown to have a negative effect on psychological mindfulness, which in turn had a negative impact upon the psychopathologies of participants. Since mindfulness is an emotion-regulating tool [11], our findings indicate that psychological mindfulness can be an indicator of the degree to which a person feels aware and accepting his or her experiences without bias judgments and thus serves as a buffer on psychopathologies. Additionally, dysfunctional coping was shown to have a positive effect on psychological inflexibility and psychological mindedness, which in turn have a positive effect on psychopathologies, probably because psychological inflexibility represents the opposite emotional states: disconnected with real sensations, feelings, or thoughts. Therefore, individuals in turn develop greater psychopathologies. However, an unexpected finding was that dysfunctional coping had a positive effect on psychological mindedness, which in turn has a positive effect on psychopathologies. Nevertheless, Hansell et al. [44] reported that psychological mindedness was positively correlated with psychopathologies. Due to the fact that highly psychologically minded individuals are more attuned than others to inner contradictions and dilemma, therefore, individuals are more likely to feel guilty and insecure [45]. In this sense, highly psychologically minded individuals are highly in introspective but also miserable. Essentially, the findings of this study suggest that the benefits of psychological mindedness during the COVID-19 pandemic are inextricably linked to its costs: greater psychopathologies.

This is a crucial finding as it suggests that if specific interventions that target psychological mindedness, psychological mindfulness, and psychological flexibility are performed, which are crucial components of Acceptance and Commitment Therapy (ACT), and we increase the levels of both constructs, clinically, it is implied that dysfunctional coping mechanisms will then no longer lead to psychopathologies, as the effect of the improved mediators reduces that relationship to non-significance. This concurs well with ACT theoretical models that suggest that instead of attempting to eliminate psychopathologies, which are unrealistic targets, the focus of therapy should be to learn to live with the distress better using psychological mindedness and mindfulness skills and to be more psychologically flexible so that one can live a more meaningful and value-driven life despite the pain. This is especially prescient in COVID-19, where the trigger is unseen, potentially carried by asymptomatic carriers, untreatable, and not vaccinated at the point of the study being performed. Hence, as the source of the distress clearly cannot be ameliorated, is worsening in certain countries, and is unlikely to go away anytime soon due to the lack of treatment and vaccine, this study supports wholeheartedly the assertion that it is better and more efficacious to be kind to oneself and to have more skills in the armamentarium to be more psychologically flexible in order to ride through and weather the COVID-19 storm.

The alarming COVID-19 situation in Indonesia had certainly caused significant psychological distress, as evidenced by higher anxiety and stress levels among the Indonesian respondents compared to Malaysian respondents. This corroborates with findings of other studies that proposed that there is a linear relationship between the number of cases and psychological distress suffered, especially those in high-risk areas [46,47]. Although this may reflect a normal physiological reaction, however, with the number of positive cases and total deaths steadily increasing, authorities should consider large-scale mental health intervention to address and alleviate the trepidation suffered by the Indonesians and Malaysians.

Stress is the most prominent psychopathologies among the students, looking into the findings. More intervention on alleviating stress symptoms among students should be refined, and this study corroborates overwhelming evidence that show that psychological mindedness, psychological mindfulness, and psychological flexibility are efficacious in dealing with psychopathology symptoms. These three psychological processes can aid in responding to the unpredictable and untenable stress of COVID-19, with awareness of physical and emotional reactions in the present moment, rather than simply acting instinctively without an awareness of motives. This allows more adaptive reactions to difficult situations, of which the dynamic and flexible COVID-19 situation is most certainly one. By practicing psychological mindfulness, individuals can learn to maintain a relaxed mind and concentrate on the present moment [48]. Hence, this study’s important clinical implication is that psychosocial interventions and also needs assessments should be inclined towards risk groups; i.e., females and those in Indonesia, and should be grounded in mindedness, mindfulness, and flexibility-oriented psychological approaches.

### Limitation and Implication

There are a few limitations of this study. Firstly, it is a cross sectional study, and hence causal inferences cannot be made. Secondly, due to the dearth of available Indonesian language questionnaires for psychological process variables, validated Malay questionnaires were used and adapted with Indonesian variations to accommodate for UNUSA respondents. Due to the urgency and dynamic requirements of the COVID-19 pandemic, there was insufficient time to perform a formal validation study on the adapted Indonesian questionnaires. As the Malay and Indonesian language are largely mutually intelligible, the face validity of the Malay versions was confirmed with Indonesian researchers in this study; but for future research purposes, the validation of all three psychological process variables into Bahasa Indonesia are in progress by the same teams. Thirdly, the survey was performed through online methods due to social distancing and lockdowns; hence there may be higher levels of social desirability.

Nevertheless, quarantine orders implemented by both countries most certainly will create difficulties in implementing psychological interventions. However, creative approaches have been encouraged and implemented. Online counselling has evolved in COVID-19 to be more approachable, private, secure, and more easily accessed. Mindfulness training especially has started to evolve into video- or audio-guided home practice daily with specialised trainers in small groups. For psychological flexibility, online Acceptance and Commitment Therapy (ACT) programs, with their focus on discrete skill building, can supplant face-to-face approaches. It is a constructive method to shape behavior with a target to enhance the quality of life, with an aim to support clients to adopt life goal-directed behaviors to manage anxiety [48].

In conclusion, this study has provided an insight on the importance of various psychological process students’ psychopathologies during the COVID-19 pandemic. In light of this, factors such as psychological mindedness, psychological mindfulness, and psychological flexibility are important. The high level of stress and anxiety among the Indonesian sample compared to the Malaysian sample in this study showed that most Indonesian students are uniquely susceptible to mental health difficulties during COVID-19. Hence, this study aspires to develop a more refined and polished intervention especially for students in Malaysian and Indonesian public universities who are subjected to the quarantine order as implemented by both government authorities.

## Figures and Tables

**Figure 1 behavsci-12-00206-f001:**
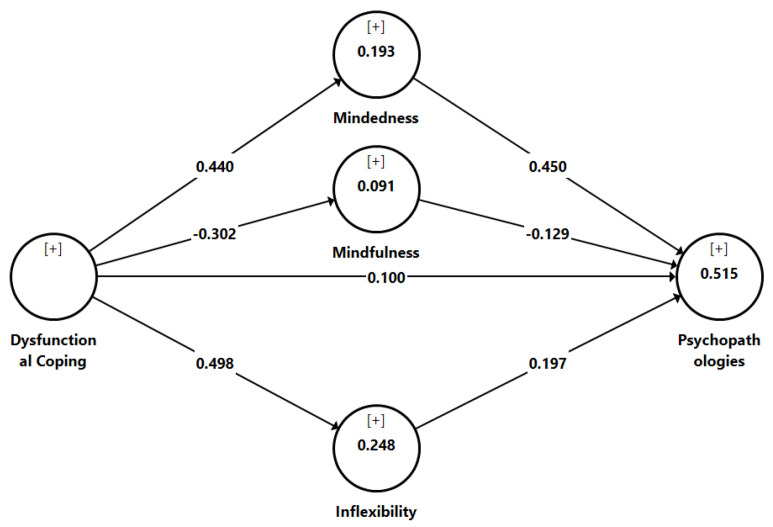
Results of structural model.

**Table 1 behavsci-12-00206-t001:** Results of measurement model.

Construct	Items	Loadings	CR	AVE	Mean	SD
Psychopathologies	A4	0.655	0.920	0.511	1.93	0.65
	A5	0.713				
	A7	0.650				
	D3	0.755				
	D4	0.790				
	D5	0.674				
	D6	0.690				
	S4	0.748				
	S5	0.797				
	S6	0.690				
	S7	0.684				
Dysfunctional Coping	DC4	0.796	0.805	0.582	1.97	0.64
	DC5	0.658				
	DC7	0.824				
Psychological Mindfulness	MAAS1	0.701	0.904	0.513	3.84	1.03
	MAAS10	0.705				
	MAAS14	0.74				
	MAAS2	0.718				
	MAAS3	0.772				
	MAAS4	0.709				
	MAAS5	0.617				
	MAAS8	0.733				
	MAAS9	0.739				
Psychological Mindedness	INSIGHT1	0.769	0.899	0.504	2.39	0.88
	INSIGHT2	0.587				
	INSIGHT3	0.724				
	INSIGHT4	0.812				
	INSIGHT5	0.799				
	INSIGHT6	0.716				
	INTEREST1	0.522				
	INTEREST3	0.546				
	INTEREST5	0.834				
Psychological Inflexibility	AAQ1	0.768	0.911	0.594	3.35	1.39
	AAQ2	0.835				
	AAQ3	0.851				
	AAQ4	0.773				
	AAQ5	0.733				
	AAQ6	0.695				
	AAQ7	0.729				

**Table 2 behavsci-12-00206-t002:** Discriminant validity: Fornell–Larcker.

No	Variables	1	2	3	4	5
1	Dysfunctional Coping	**0.763**				
2	Inflexibility	0.498	**0.771**			
3	Mindedness	0.440	0.621	**0.710**		
4	Mindfulness	−0.302	−0.364	−0.436	**0.716**	
5	Psychopathologies	0.436	0.574	0.673	−0.428	**0.715**

*Note.* Square root of the AVE for each construct (bold) was higher than its correlation with other constructs; discriminant validity was therefore established.

**Table 3 behavsci-12-00206-t003:** Descriptive analysis of respondents.

		Indonesian (N = 869)			Malaysian (N = 515)		
		Mean	Frequency	Percent	Mean	Frequency	Percent
Age		20.5			22.2		
Gender	Male		133	15.3		158	30.7
	Female		736	84.7		357	69.3
Depression	Normal		530	61.0		156	30.3
	Mild		146	16.8		196	38.1
	Moderate		98	11.3		142	27.6
	Severe		72	8.3		18	3.5
	Extra Severe		23	2.6		3	0.6
Anxiety	Normal		402	46.3		171	33.2
	Mild		163	18.8		215	41.7
	Moderate		141	16.2		118	22.9
	Severe		73	8.4		11	2.1
	Extra Severe		90	10.4		0	0.0
Stress	Normal		561	64.6		126	24.5
	Mild		104	12.0		158	30.7
	Moderate		141	16.2		208	40.4
	Severe		48	5.5		20	3.9
	Extra Severe		15	1.7		3	0.6

**Table 4 behavsci-12-00206-t004:** Results of direct and indirect effect.

Relationship	*β*	*t*	*p*	∆R^2^	VIF	Supported
Dysfunctional Coping → Psychological Inflexibility	0.498	16.987	0.00	0.248	1.000	Yes
Dysfunctional Coping → Psychological Mindedness	0.440	13.948	0.00	0.193	1.000	Yes
Dysfunctional Coping → Psychological Mindfulness	−0.302	8.253	0.00	0.091	1.000	Yes
Dysfunctional Coping → Psychopathologies	0.100	3.492	0.00	0.515	1.395	Yes
Psychological Inflexibility → Psychopathologies	0.197	7.598	0.00		1.658	Yes
Psychological Mindedness → Psychopathologies	0.450	14.512	0.00		1.776	Yes
Psychological Mindfulness → Psychopathologies	−0.129	4.966	0.00		1.257	Yes
Dysfunctional Coping → Psychological Inflexibility → Psychopathologies	0.117	6.658	0.00			Yes
Dysfunctional Coping → Psychological Mindfulness → Psychopathologies	0.042	4.068	0.00			Yes
Dysfunctional Coping → Psychological Mindedness → Psychopathologies	0.206	9.634	0.00			Yes

**Table 5 behavsci-12-00206-t005:** Mean scores for anxiety, depression, and stress for Indonesia and Malaysia.

Variables	Indonesia		Malaysia	
	Mean	SD	Mean	SD
Anxiety	1.78 **	0.55	1.64 **	0.52
Depression	1.67	0.59	1.72	0.60
Stress	1.93 **	0.63	1.82 **	0.59

Note: ** *p* < 0.01.

## Data Availability

Data can be available upon reasonable request.

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
