# Peer review of "Mediating Effect of Psychological Process Variables on the Relationship between Dysfunctional Coping and Psychopathologies: A Comparative Study on Psychopathologies during COVID-19"

_behavsci, 2022, doi:10.3390/bs12070206_

Round 1

Reviewer 1 Report

This paper aims to identify the mediating role of psychological process variables namely, psychological mindedness, psychological mindfulness, and psychological inflexibility on the relationship between dysfunctional coping and psychopathologies in Indonesian undergraduate students who are subjected to national quarantine orders; and to compare the level of anxiety, depression, and anxiety between Indonesian and Malaysian undergraduate students. 

I believe this study doesn't present a novelty and could be a report study with statistical point of view. 

The paper is well written and the sections are well connected, the abstract is supported by the conclusion. 

I have added some remarks and questions in the attached document!

Author Response

Dear Examiner,

We are grateful for your consideration of this manuscript, and we also very much appreciate your suggestions, which have been very helpful in improving the manuscript. All the comments we received on this manuscript have been taken into account in improving the quality

Reviewer 2 Report

The study is interesting for its content and for the methodological rigour employed. I share the importance of promoting the appropriate use of mental and coping skills to manage problems that affect mental health. Personally I prefer to use the term Mental Health rather than Psychopathologies.

I would like to know what made you choose to use structural equations instead of other techniques, such as multiple regression models that include interactions between variables.

When you state: the study showed that psychological mindedness, psychological mindfulness, and psychological inflexibility significantly mediate the relationship between dysfunctional coping and psychopathologies. I personally suggest, for a better understanding by the readers, to explain the contribution of each of these constructs on the mental health of the subjects. That is to say, to comment more on their role as risk factors or as protective factors.

In this sense, I think it is worth commenting a little more on the role of mindfulness.

On the other hand, a much higher percentage of women than men is observed in both samples. Perhaps it would also be interesting to assess gender differences in the different analyses.

The limitations of the study already indicate that, in accordance with the type of design used, it is not advisable to speak of causal relationships. However, I am left with the question of whether dysfunctional coping styles are antecedent variables ("possible causes") or whether they are a consequence.

Despite the drawbacks of adapting some of the questionnaires, the study as a whole is interesting and reflects quality in its approach. Congratulations for the work.

Author Response

(The authors gave the same response as above.)
